# Deletion of MGF505-2R Gene Activates the cGAS-STING Pathway Leading to Attenuation and Protection against Virulent African Swine Fever Virus

**DOI:** 10.3390/vaccines12040407

**Published:** 2024-04-11

**Authors:** Sun-Young Sunwoo, Raquel García-Belmonte, Marek Walczak, Gonzalo Vigara-Astillero, Dae-Min Kim, Krzesimir Szymankiewicz, Maciej Kochanowski, Lihong Liu, Dongseob Tark, Katarzyna Podgórska, Yolanda Revilla, Daniel Pérez-Núñez

**Affiliations:** 1Careside Co., Ltd., Sagimakgol-ro 45 Beongil 14, Seongnam-si 13209, Gyeonggi-do, Republic of Korea; sunwoosy@gmail.com; 2Microbes in Health and Welfare Department, Centro de Biologia Molecular Severo Ochoa (CBM), CSIC-UAM, c/Nicolás Cabrera 1, 28049 Madrid, Spain; raquel.g.b@cbm.csic.es (R.G.-B.); gvigara@cbm.csic.es (G.V.-A.); 3Department of Swine Diseases, National Veterinary Research Institute, 57 Partyzantów Avenue, 24-100 Pulawy, Poland; marek.walczak@piwet.pulawy.pl (M.W.); krzesimir.szymankiewicz@piwet.pulawy.pl (K.S.); maciej.kochanowski@piwet.pulawy.pl (M.K.); katarzyna.podgorska@piwet.pulawy.pl (K.P.); 4Laboratory for Infectious Disease Prevention, Korea Zoonosis Research Institute, Jeonbuk National University, 79 Gobong-ro, Ma-dong, Iksan 54531, Jeollabuk-do, Republic of Korea; daeminkk@gmail.com (D.-M.K.); tarkds@jbnu.ac.kr (D.T.); 5Department of Microbiology, Swedish Veterinary Agency, 751 89 Uppsala, Sweden; lihong.liu@sva.se

**Keywords:** ASFV, Arm/07/CBM/c2, MGF505-2R, cGAS-STING, IFN-β, LAV, vaccine

## Abstract

African swine fever virus (ASFV) is the etiological agent causing African swine fever (ASF), affecting domestic pigs and wild boar, which is currently the biggest animal epidemic in the world and a major threat to the swine sector. At present, some safety concerns about using LAVs against ASFV still exist despite a commercial vaccine licensed in Vietnam. Therefore, the efforts to identify virulence factors and their mechanisms, as well as to generate new vaccine prototypes, are of major interest. In this work, we have identified the MGF505-2R gene product as an inhibitor of the cGAS/STING pathway, specifically through its interaction with STING protein, controlling IFN-β production. In addition, immunization of a recombinant virus lacking this gene, Arm/07-ΔMGF505-2R, resulted in complete attenuation, demonstrating its involvement in ASFV virulence. Finally, immunization with Arm/07-ΔMGF505-2R induced the generation of antibodies and proved to be partially protective against virulent ASFV strains. These results identify MGF505-2R, as well as its mechanism of action, as a gene contributing to understanding the molecular mechanisms of ASFV virulence, which will be of great value in the design of future vaccine prototypes.

## 1. Introduction

African swine fever virus (ASFV), a dsDNA virus belonging to the Asfarviridae family [1], is the etiological agent of African swine fever (ASF), a severe disease affecting both wild boar and domestic pigs, whose virulent strains reach up to 100% mortality. ASFV was first identified in Africa in 1921 and spread to the Caucasus region in 2007, spreading through Russia and East Europe, affecting many countries such as Romania, Poland, Latvia, Hungary, and Czech Republic, and more recently reaching Western European countries, including Belgium, Italy, and Germany [2]. Currently, it is endemic in China, the world’s largest pork producer, where it was first detected in 2018 [3], and presently affects neighboring countries such as Vietnam, Laos, Myanmar, Korea, the Philippines, Indonesia, and India [4,5,6]. It emerged in 2021 for the first time in many decades in the Americas, namely in the Dominican Republic and Haiti [7]. Subsequently, the virus has continued to spread, and since January 2021, ASF has been reported in five different world regions, affecting more than 40 countries in Europe, Oceania, and Asia [8]. The widespread emergence of ASFV in major pork-producing countries has resulted in a destabilization of the food chain and is one of the most important socio-economic and industrial animal health concerns worldwide. The development of novel, safe, and efficacious vaccines is urgently needed to address the threat to worldwide pork production.

An important issue in the field of ASFV is the identification of the molecular factors that determine its virulence, which can be up to 100% in virulent strains in relation to other naturally attenuated strains that only cause chronic disease [9,10]. In this regard, our group described that the cGAS/STING pathway is only efficiently controlled by virulent ASFV strains, while attenuated ASFV strain infection leads to IFN-β production [11]. Subsequent studies have reinforced this finding, identifying a multitude of viral factors that control some of the elements of the cGAS/STING pathway [12,13,14,15,16,17,18,19]. The type I IFN-induced JAK/STAT pathway has also been described as being controlled by ASFV [20], further relating type I IFN to virulence. Some of the viral factors that control this pathway during ASFV infection [21,22,23,24,25] have also been identified. The identification and study of these factors are fundamental for the development of live attenuated vaccines (LAVs), which are currently the most realistic strategy for developing a vaccine against ASFV.

Several LAV candidates for ASF have been developed in recent years, showing different degrees of protection, from 60 to 100%, which presented different side effect profiles in vaccinated animals [26,27,28,29,30,31]. More recently, other vaccine prototypes have been developed that achieve 100% protection and apparently do not present safety problems in vaccinated animals in these studies [32,33,34]. In fact, one of these candidates might be used as a vaccine in Vietnam, although it is still under study [35].

In the present work, we have identified the pMGF505-2R protein as an inhibitory factor of the cGAS/STING pathway. This protein contains a LxCxE domain that is found in E7 from human papillomavirus (HPV) and E1A from adenovirus, which binds the STING protein and inhibits the cGAS/STING pathway through this motif [36]. Ectopic expression studies have shown, on the one hand, that pMGF505-2R is able to inhibit TBK-I phosphorylation as well as IFN-β production upon activation of the cGAS/STING pathway and, on the other hand, that it is able to interact with STING. In addition, we have generated a recombinant virus derived from the virulent genotype II Arm/07/CBM/c2 ASFV [37] in which the MGF505-2R gene has been deleted (Arm/07-ΔMGF505-2R-GFP) by using CRISPR-Cas9 technology in COS-1 cells, which is a cell line where other LAVs have been generated previously [34,38] and that would be suitable for industrial scale-up.

In comparison with the parental Arm/07/CBM/c2 virus, studies with the Arm/07-ΔMGF505-2R-GFP recombinant virus indicated that it failed to control the activation of the cGAS/STING pathway and consequently the IFN-β production, confirming the role of pMGF505-2R as an innate—response modulating factor. Furthermore, two separate in vivo studies performed at different doses (10^2^ and 10^3^ TCID50) demonstrated complete attenuation of the recombinant virus, thus connecting the control of the cGAS/STING pathway and type I IFN with ASFV virulence. Finally, vaccination studies have demonstrated the ability of the Arm/07-ΔMGF505-2R-GFP prototype to induce protection of up to 75% of vaccinated animals against different virulent challenges with circulating strains. This study places pMGF505-2R protein as a key player in the control of the cGAS/STING pathway and in ASFV virulence, contributing to further studies aimed at the generation of new safe and effective vaccines against ASFV.

## 2. Materials and Methods

### 2.1. Cells and Viruses

Porcine alveolar macrophages (PAMs) were extracted from bronchoalveolar lavage and washed with PBS as described in [39] and cultured in Dulbecco-modified Eagle medium (DMEM) supplemented with L-glutamine (2 mM), gentamicin (100 U/mL), and 10% porcine serum. Cercopithecus aethiops kidney cells (COS-1 cells) and human embryonic kidney cells (HEK-293T cells) were obtained from the American Type Culture Collection (ATCC). BSRT7/5 cells, which are Baby Hamster Kidney cells (BHK-21) cells, constitutively expressing T7 phage RNA polymerase, were kindly provided by the laboratory of Dr. Adolfo Garcia Sastre (Icahn School of Medicine at Mount Sinai, New York, NY, USA). COS-1, BSRT7/5 and HEK-293T cells were grown in DMEM, supplemented with L-glutamine (2 mM), gentamicin (100 U/mL), and 5% fetal bovine serum (FBS; Invitrogen Life Technologies, Carlsbad, CA, USA).

Viruses used were the ASFV viral strains Arm/07/CBM/C2 (LR812933.1) [37], NH/P68 (NC_044943.1), the recombinant virus Arm/07-ΔMGF505-2R-GFP generated in the present work, and virulent ASFV Korean isolate ASF/Korea/Pig/Paju/2019 (MTT748042.1) [34] (all of which were produced in PAMs), and Modified Vaccinia Ankara Virus (MVA) encoding for the T7 RNA polymerase of bacteriophage T7 (MVA-T7), which was produced in BHK-21 cells [40]. In vitro infections were performed with previous adsorption by incubating on ¼ of the total volume of the plate at 37 °C for 90 min. Then, it was removed, and fresh medium was added to the cells and incubated at 37 °C for the time indicated in each experimental condition.

### 2.2. Antibodies for In Vitro Experiments

The polyclonal antibody anti-STING (19851-1-AP) was purchased from Proteintech, Rosemont, IL, USA. Monoclonal antibodies anti-pTBK1 E8I3G (38066), anti-Myc (clone 9B11) (2276), anti-Flag-Tag DYKDDDDK (2368), anti-pSTING Ser366 (85753), and anti-p-IRF3 Ser396 (4947) were purchased by Cell Signaling, Danvers, MS, USA. Monoclonal antibodies anti-TBK1 (sc-73115) and anti-actin (C-4) (Sc-47778) were purchased by Santa Cruz, Santa Cruz, CA, USA. Monoclonal antibody anti-tubulin (B-5-1-2) (T-5168) was purchased by Sigma, Kanagawa, Japan. Polyclonal antibody anti-I125L was previously generated in our lab [24]. Monoclonal antibody anti-p32(S-5C1) was previously generated [34]. Anti-rabbit (NA934VS) and anti-mouse (NXA931V) immunoglobulin G coupled to peroxidase antibodies were purchased from Amersham Biosciences, Buckinghamshire, UK.

### 2.3. Vectors and Transfection

For the luciferase assay, pCAGGS empty vector (chicken β-actin promoter), pIFNβ_luc vector (coding for luciferase protein under the IFNβ promoter), and the pRLTK vector (containing the Renilla luciferase reporter gene under the Herpes simplex thymidylate kinase promoter) were kindly provided by Dr. Adolfo García-Sastre (Icahn School of Medicine at Mount Sinai, NY, USA). For transient MGF505-2Rmyc expression, we cloned the MGF505-2R gene from the Arm/07/CBM/c2 strain (GenBank: LR812933.1) together with myc epitope into a pCAGGS empty vector. Previously, we generated pIRES_MGF505-2R-myc that was used as a template. For pcDNA_MGF505-2R-myc-His cloning, we used 2X Phusion Maser Mix HF (Thermo Scientific, Waltham, MA, USA) and In-Fusion technology (Takara), San Jose, CA, USA by using the following oligo probes: 5′-GAACAAAAACTCATCTCAGAAGAGGATCT-3, 5′-GAACCGCGGGCCCT-3′; and 5′-GAGGGCCCGCGGTTCATGTTTTCCCTTCAAGACCTTTGC-3′, 5′-GATGAGTTTTTGTTCACATTGTATTGTTTTATAATGAATTACTTTATATAATATTTCTAAAC-3′ for insert amplification from viral DNA. For pIRES-MGF505-2R-myc cloning, we used the previously generated pcDNA_MGF505-2R-myc-His vector as a template to clone into a pIRES vector, using the same technology with the following oligo probes: 5′-TAATTCTAGATCATCGAAACATGAGGATCACCCATa-3′ and 5′-GGATCCTCGAGCTCAGGGT-3′; and 5′-TGA GCT CGA GGA TCC ATG TTT TCC CTT CAA GAC CTT TGC C-3′, 5′-GATGATCTAGAATTACAGATCCTCTTCTGAGATGAGTTT-3′. For pCAGGS_IRES_MGF505-2Rmyc cloning, 4 µg of pCAGGS empty vector was digested with EcoRI-HF and BglII (New England Biolabs, Ipswich, MA, USA) followed by incubation with 2 µL of calf intestinal alkaline phosphatase (CIP) (New England BioLabs). The insert from pIRES-MGF505-2R-myc was amplified with PCR using 2X Phusion Master Mix HF (Thermo Scientific) with the following oligo probes: 5′-CTAGAATTCAGCAGGTTT-3′ and 5′-GCCAGATCTTTACAGATCCTCTTCTG-3′. Digestion products were incubated with T4 ligase (Promega) for 15 min at room temperature.

For the co-immunoprecipitation assay, the STING_Flag vector was kindly provided by Dr. Adolfo García-Sastre (Mount Sinai, NY, USA).

For the recombinant virus generation using CRISPR/Cas9, we generated the following vectors: pFL_MGF505-2R, pFLΔMGF505-2R_p72-GFP and pSpCas9(BB)ΔNLS-2A-Puro_MGF505-2R-gRNA-15 and -30. For pFL_MGF505-2R and pFLΔMGF505-2R_p72-GFP cloning, we used In-Fusion technology (Takara) by using the following oligo probes: 5′-CTTCTGAGGCGGAAAGAACCA-3′, 5′-AACGCGTATATCTGGCCCG-3′, 5′-CCAGATATACGCGTTACACGGCAGGAACACGTATAAGC-3′, 5′-TTTCCGCCTCAGAAGGAATAAGCATTTCAGTGAACAGGTG-3′ for pFL_MGF505-2R vector; and 5′-AATTCAATAGATATCCATCATTAATATTGATTATATTTTCGA-3′, 5′-ATCCCCCTACTTCATTAAAAAATAAAAATTCTAGCT-3′, 5′-ATGAAGTAGGGGGATTATTTAATAAAAACAATAAATTATTTTTATAACATTATATAGGTCGCCAC-3′, 5′-GATATCTATTGAATTCCATAGAGCCCACCGCA-3′ for pFLΔMGF505-2R_p72-GFP cloning. For pSpCas9(BB)ΔNLS-2A-Puro vectors, cloning was performed using the technology described in [41], including gRNA sequence 5′-TCGTACTCTATCACATAGCG-3′ (gRNA-15) or 5′-GCTCTATGTTGCTCAAAACG-3′ (gRNA-30), respectively.

For the transfection protocol, FuGene HD transfection reagent (Promega, Madison, WI, USA) was employed, following the manufacturer’s instructions.

### 2.4. Luciferase Assay

HEK-293T cells were seeded on M24 plates (1.5·10^5^ cells/well) and co-transfected with the plasmids pIFNβ-luc (50 ng/well) and Renilla luciferase reporter construct pRLTK (25 ng/well) together with TBK1_Flag (50 ng/well) and either pCAGGS_IRES_MGF505-2Rmyc (1000 ng/well) plasmids or empty vector. At 24 h post-transfection (hpt), the cells were collected and processed as indicated by the manufacturer of the Luc-PairTM DuoLuciferase HS kit (GeneCopoeia). Finally, luciferase readings were performed using the FLUOstar OPTIMA reader (BMG LabTech, Ortenberg, Germany).

### 2.5. Western Blot

At each indicated condition, cells were washed with PBS and resuspended in RIPA lysis buffer (50 mM Tris-HCl pH = 7.4, 150 mM NaCl, 1% Triton, 0.5% Sodium Deoxycholate, 0.1% SDS) supplemented with protease and phosphatase inhibitors (Roche, Basel Switzerland). The extracts were kept 30 min at 4 °C and then centrifuged 5 min at 14,000 rpm to remove cell debris. Finally, total protein concentration was determined using the bicinchoninic acid (BCA) Protein Assay Kit (Pierce, Appleton, WI, USA) spectrophotometric quantification method according to the manufacturer’s instructions.

Equal amounts of total protein from each sample, to which 5× Leammli loading buffer (1M Tris buffer pH = 6.8, 10% SDS, 4% β-Mercaptoethanol, 20% Glycerol and 0.01% Bromophenol Blue, Solon, Ohio, USA) was added and incubated at 95 °C for 5 min. Proteins were separated by electrophoresis on polyacrylamide-SDS gels (SDS-PAGE) and transferred to Immobilon-P membranes (Millipore, Burlington, MA, USA). Next, the membranes were incubated with the indicated primary antibodies diluted in blocking solution (Tris Saline buffer (TBS) with 1% skimmed milk powder), after which the membranes were washed with TBS three times for 5 min, followed by incubation with the corresponding peroxidase-coupled secondary antibody according to the species of the first antibody. Finally, the identification of specific protein bands was performed using the ECL Western Blotting Analysis System chemiluminescence method (Amersham Pharmacia, Peapack, NJ, USA).

### 2.6. Co-Immunoprecipitation Assay

COS-1 cells were transfected with STING-Flag (0.5 µg/1·10^6^ cells) and either pCAGGS_IRES_MGF505-2Rmyc or an empty vector (3 µg/1·10^6^ cells). After 24 hpt, the cells were harvested and lysed following the manufacturer’s recommendations using the PierceTM Classic Magnetic IP/Co-IP kit (ThermoFisher). Briefly, the cells were lysed in 500 µL of lysis buffer supplemented with protease and phosphatase inhibitors (Roche) and incubated for 2 h on ice. The lysates were then centrifuged at 13,000 for 5 min at 4 °C to remove cell debris. A portion of the lysates was used as total cell extract “Input”, and the remainder was used for immunoprecipitation. The A/G magnetic beads were bound to the anti-STING antibody (1:50 dilution) and were incubated with the rest of the lysate under agitation for 16 h at 4 °C. Then, beads were washed, and the proteins were eluted with elution buffer. Subsequently, immunoprecipitated and input samples were analyzed using Western blot.

### 2.7. Generation and Isolation of Arm/07-ΔMGF505-2R-GFP Virus by CRISPR/Cas9 Technology

COS-1 cells were seeded on a six-well plate at 90% confluence and co-transfected with specific pSpCas9(BB)ΔNLS-2A-Puro gRNA (2 µg) and pFLΔMGF505-2R_p72-GFP vector (2 µg) with FuGene HD (Promega, Madison, WI, USA). After 24 h, puromycin at 1 µg/mL (Sigma, Saint Louis, MO, USA) was added to transfected cells. A total of 48 h after the addition of puromycin, transfected cells were infected with the Arm/07/CBM/c2 ASFV strain at two different MOI (1 and 0.1) to generate Arm/07-ΔMGF505-2R-GFP. Five days post-infection (dpi), cells and medium were collected and conserved at −80 °C.

This virus-containing sample was used to infect COS-1 cells. After adsorption (1 h 30 min), the inoculum was removed, and carboxymethylcellulose (Sigma, Saint Louis, MO, USA) in DMEM 2% fetal bovine serum was added. When viral plaques appeared and were identified with fluorescent microscopy (4–7 dpi), they were collected with sterile tips in 40 µL of DMEM and preserved at −80 °C. The extracted virus, after three freeze–thaw cycles, was used to infect COS-1 cells, repeating the same procedure until the recombinant virus was isolated from the parental virus. Isolation of the recombinant virus from the parental virus was checked with PCR. For that, 10 µL of the isolated plaque was digested with proteinase K (Sigma, Saint Louis, MO, USA) in 1.5 mM MgCl_2_, 50 mM KCl, 0.45% Tween20, 0.45% NP40, and 10 mM Tris-HCl pH 8.3 buffer, incubated for 30 min at 45 °C, then incubated for 15 min at 95 °C. To detect the presence of the recombinant and parental viruses, the digested isolated plaque was used as a DNA template for specific PCR. The oligos used were 5′-TTTATCTTACCATTGATTCAAGACGCGA-3′ and 5′-TAGCGCGGTAGTTTTTTGATGC-3′ for the detection of MGF505-2R; and 5′-ACATGG TCCTGCTGGAGTTC-3′ and 5′-GATATCTATTGAATTCCATAGAGCCCACCGCA-3′ for the detection of GFP.

After the recombinant virus was isolated, it was grown to obtain a viral stock. For that, eight to ten P150 plates of COS-1 cells were infected during 3 days, when the total virus was collected and subjected to freeze–thaw cycles. After centrifugation (3000 rpm 5 min at RT), supernatant was then collected and again centrifuged (7000 rpm o/n at 4 °C). The resulted pellet was finally resuspended in fresh DMEM medium.

### 2.8. Viral DNA Extraction for NGS Analysis

Recombinant Arm/07-ΔMGF505-2R-GFP virus was amplified on PAMs (1.6 × 10^7^) and at 3 dpi, viral DNA from extracellular particles was extracted as previously indicated [37]. Briefly, after a first centrifugation where the supernatant containing the extracellular particles was collected, it was centrifuged at 8281× *g* for 16 h at 4 °C, and the pellet obtained was resuspended in cold, filtered 10 mM Tris-HCl (pH = 8.8). It was then treated with DNAase I (Sigma) 0.25 U/mL, Nuclease S7 (Sigma) 0.25 U/mL, and RNAase A (Promega) 20 µg/mL in nuclease buffer (0.8 M Tris-HCl, pH = 7.5, 0.2 M NaCl, 20 mM CaCl_2_, 120 mM MgCl2) and incubated for 2 h at 37 °C. Next, 12 mM EDTA pH = 8 and 2 mM EGTA were added, incubated for 10 min at 75 °C and treated with proteinase K (Sigma) at 200 µg/mL in 0.5% SDS for 1 h at 45 °C. For viral DNA precipitation, it was incubated 1:1 with phenol:chloroform:isoamyl alcohol (25:24:1) and centrifugated (9400× *g*, 3 min at RT). The aqueous fraction, 0.1 volumes of 3 M acetic acid (pH = 5.2), 2 volumes of cold 100% ethanol and 1 µL of LPA (Sigma) were mixed and incubated at −80 °C o/n. Then, the sample was centrifugated (15,890× *g*, 30 min at 4 °C), and the supernatant was discarded. Finally, the pellet was washed with cold 70% ethanol, air dried, and finally resuspended in 10 mM Tris (pH = 8.8).

### 2.9. Illumina Sequencing and Data Analysis

Recombinant Arm/07-ΔMGF505-2R-GFP virus DNA was sequenced with Illumina technology at MicrobesNG (Birmingham, UK) using MiSeq. Reads were analyzed as described in [37]. Briefly, reads were trimmed with trimmomatic software [42] v0.39), and quality analysis was performed with FastQC software (v0.11.8). Reads were aligned to the Arm/07/CBM/c2 parental genome (LR812933.1) using samtools (v1.7), and indexed BAM files were visualized with IGV software (Integrative Genomics Viewer, v2.8.6) using the Arm/07/CBM/c2 genome as reference. For variant analysis, GATK software (v4.1.2) was used with standard parameters, and variants were mapped to Arm/07/CBM/c2 ORFs using SNPEff software (v4.3t) [43]. The modifications that appeared in the genome of the recombinant virus Arm/07-ΔMGF505-2R-GFP after sequencing by Illumina were confirmed by Sanger (Macrogen, Seoul, Korea). For this purpose, the specific regions were amplified using the following primers: MGF110-14L: 5′-TCC GTG ATT CCA AGA CTC AA-3′, 5′-CTG TAG GCT GAA AAC AAT CCA TAT AAT G-3′; MGF505-7R: 5′-GAA CTT GTT GCT ATC TTA CAT AAA TTA CAA G-3′, 5′-GTA AGA ATC TCA AGC TTC CGA TTT TC-3′; EP424R: 5′-CTA TGG TCT TTA TCA GTG CAA TCC-3′, 5′-GTT ATC CCA ATT ATG ATG CTT AAT GG-3′; CP530R: 5′-GCC CTC TAA TAT GAA ACA GTT TTG-3′, 5′-GTA ATC TGG GTA AAA AAC TTT TTA AAC TCC AG-3′.

A coverage plot of Arm/07-ΔMGF505-2R-GFP reads against Arm/07/CBM/c2 parental genome (LR812933.1) was generated using bedtools (v2.27.1) for coverage values calculation and Rstudio software (v2022.02.3+492) for histogram generation using plot function.

### 2.10. Viral Growth Kinetics

PAMs were seeded at a concentration of 1.5·10^6^ cells/well and mock-infected or infected with either Arm07/CBM/c2 (WT) or Arm/07-ΔMGF505-2R-GFP at a multiplicity of infection (MOI) = 0.5, in DMEM–10% porcine serum. Cells were then collected at 0, 24, 48, or 72 hpi and subjected to freeze–thaw cycles (×3). For quantification, titration was performed by a Reed & Muench TCID-50 assay [44] in COS-1 cells. Briefly, several dilutions of each sample were used to infect COS-1 cells s (7000 cells/well) and 72 hpi. They were fixed with PFA 4%, and permeabilized with 0.2% Triton X-100 (in the case of Arm/07/CBM/c2). TCID50 titration assay was performed by staining viral p32 (with anti-p32 primary antibody (S-5C1) (1/500) incubation followed by anti-mouse Alexa Fluor 488 secondary antibody (A-21206) incubation, from Invitrogen) for Arm/07/CBM/c2or using green (GFP) fluorescence (in the case of Arm/07-ΔMGF505-2R-GFP). Biological duplicates were used.

### 2.11. RT-qPCR Assay

The 6·10^6^ PAMs were seeded on p60 plates and infected with the indicated viral strains. At 16 hpi, RNA was extracted from the cells using the RNeasy kit (Qiagen, Venlo, The Netherlands), and total RNA was quantified with the NanoDrop One (Thermo Scientific). Equivalent amounts of RNA were then retrotranscribed using the NZY first-strand cDNA kit (NZYTech), and RT-PCR was performed in technical triplicate with a cDNA amount of 12 ng/sample on ABI PRISM 7900HT SDS (Applied Biosystems, Waltham, MS, USA) thermal cycler using SYBR Green (NZYTech). Expression levels of genes of interest were normalized against 18s ribosomal RNA (rRNA) expression, and these values were relativized against the mean of the values obtained in the mock. The primers used are as follows: 18S: 5′-GGCCCGAGGTTATCTAGAGTC-3′, 5′-TCAAAACCAACCCGGTCA-3′; CP204L: 5′-AAAA ATGATAATGAAACCAATGAATG-3′, 5′-ATGAGGGCTCTTGCTCAAAC-3′; MGF505-2R: 5′-GAGTCCACCTTGGTGATAAAG-3′, 5′-CCATGATCGTCCTCACTTTC-3′; IFN-β: 5′-GTGGAACTTGATGGGCAGAT-3′, 5′-TTCCTCCTCCATGATTTCCTC-3′.

Animal Experiment I was performed at the Korea Zoonosis Research Institute, Jeonbuk National University, Republic of Korea.

### 2.12. Experimental Conditions

The animal study and experiments were performed, in compliance with the Animal Welfare Act, in Jeonbuk National University (JBNU) Institutional Biosafety Committee under IBC, Protocol #:JBNU2021-08-001-001 together with the Institutional Animal Care and Use Committee (IACUC, Protocol #:JBNU2022-025).

Eight domestic pigs (*Sus scrofa*) that were 7 weeks old, in healthy conditions, and purchased in a commercial local pig farm were used. The animals were serologically free from Porcine reproductive and respiratory syndrome virus (PRRSV), porcine circovirus2 (PCV2), Classical swine fever virus (CSFV), and Aujeszky’s disease (PRV), confirming a high level of health status. The vaccinated and non-vaccinated groups were composed of four pigs for each group; the latter were added at the challenge. For animal experiments, a BSL-3 Ag laboratory and facility was used. After 1 week of acclimatation being managed with appropriate feeding and water supply system, cleaning, and general veterinary care, each experimental group was separated into different pig isolators.

The vaccinated group was immunized via intramuscularly (IM) with 10^2^ TCID50/pig Arm/07-ΔMGF505-2R-GFP. Serum and whole blood samples were collected during the vaccination period. At 21 days after the immunization, pigs were challenged via IM with 10^2^ HAD50 of virulent ASFV, Korean isolate (ASF/Korea/Pig/Paju/2019, provided by APQA, MTT748042.1).

### 2.13. Samples Collection and Assessment of Clinical Signs

Whole blood and nasal and fecal swabs were collected on different days post-vaccination (dpv; 0, 3, 5, 7, 10, 14, 21, and 28) from vaccinated pigs and after the challenge at 0, 3, 5, 7, 10, 14, and 21 days post-challenge (dpc) from both vaccinated and control group pigs. Sera were collected at 0, 7, 10, 14, 21, and 28 dpv and 0, 5, 7, 14, and 21 dpc.

For virus detection in organ tissue samples, on euthanized (at 21 dpc) or found dead pigs after the challenge, tissues from the tonsil, lymph nodes (mandibular, superficial cervical, gastrohepatic, renal, mesenteric), spleen, heart, lung, liver, and kidney were analyzed. Evaluation and scoring of clinical signs were performed daily based on a previously established index [34,45] (Appendix A). These included body temperature, skin hemorrhage, recumbency, inappetence labored breathing, joint swelling, diarrhea, ocular discharge, urine, and vomiting. A maximum score of 40 was assigned. If the accumulative clinical score was >20, euthanasia was performed (i.e., the pig had severe clinical signs). Pigs euthanized or found dead after challenge were also subjected to gross lesion examination. Gross lesions were scored following our previous study [34] and were evaluated for body condition, integument, lung, liver, spleen, kidney, tonsil, and lymph nodes.

### 2.14. ASFV Real-Time qPCR

Whole blood and swab samples and tissues from necropsy were used for virus detection, which was determined by qPCR at different times post-vaccination and post-challenge. ASFV DNA was isolated according to the manufacturer’s procedure using a DNeasy blood and tissue kit (Qiagen). Sample volumes were 100 µL for whole blood sample volume and 200 µL for swab samples. A total of 100 µL of whole blood was mixed with 100 µL of PBS, 4 µL of RNase, 20 µL of proteinase K, and 200 µL of Buffer AL. The 200 µL swab samples were mixed with 4 µL of RNase, 20 µL of proteinase K, and 200 µL of Buffer AL. Both mixtures were incubated (56 °C, 10 min), and then 200 µL of absolute ethyl alcohol was added and mixed by vortexing. Lysate was then put into the column, centrifuged, and washed with both buffer AW1 and AW2. Then, DNA was eluted with 200 µL of Buffer AE.

For ASF-specific real-time PCR, a commercial kit (Vet maxTM African swine fever virus detection kit, ThermoFisher, Waltham, MA, USA), which was validated and certified by the WOAH, was used for ASFV detection. A total of 5 µL of sample DNA was mixed with 20 µL of reagents. The PCR conditions were as follows: 50 °C for 2 min; 95 °C for 10 min; 45 cycles of 95 °C, 15 s and 60 °C, 1 min. Results are expressed in cycle threshold (Ct), defined as the number of cycles required to cross the threshold. Ct levels are inversely proportional to the amount of target nucleic acid in the sample.

### 2.15. ELISA Assay

For humoral immune response evaluation, pig sera were collected, and ASF-specific antibodies were measured with commercial ELISA INgezim PPA COMPAC (Ingenasa, Madrid, Spain), according to the manufacturer’s recommendation. Briefly, 50 µL of serum with an equal volume of diluent was incubated for 60 min at 36 °C and washed with washing solution (×4). A total of 100 µL of conjugate was then added and incubated (30 min at 36 °C) and washed (×5). Finally, the substrate was added (100 µL) and incubated for 15 min at RT. The OD value at 450 nm was measured after incubation with 100 µL of stop solution. Two cutoffs were used to interpret the results. Samples were considered positive if their OD value was equal to or lower than the positive cutoff and negative if their OD value was equal to or higher than the negative cutoff.

Animal Experiment II was performed at the National Veterinary Research Institute Pulawy (Poland).

### 2.16. Experimental Conditions

The study was conducted according to the Polish Act of 15 January 2015 on the protection of animals used for scientific or educational purposes (Official Number: 2015/266) based on Directive 2010/63/EU of the European Parliament and of the Council of 22 September 2010 on the protection of animals used for scientific purposes. The study was approved by the Local Ethical Commission for Animal Experiments in Lublin (approval number 56/2020, 20/2023 and 21/2023).

A total of nine pigs, aged six weeks, of both sexes, were divided into two experimental groups: Group I (Arm/07-ΔMGF505-2R n = 5, pigs A–E) and Group II—Arm/07/CBM/c2 wild type (WT, control, n = 4, pigs F–I). The animals were purchased from a commercial local pig farm, with a confirmed high level of health status—serologically free from Porcine reproductive and respiratory syndrome virus (PRRSV), Aujeszky’s disease (PRV). The animals were acclimatized for 7 days in a BSL3 (Biosafety Level-3) animal facility in two independent units with permanent access to feed and water. After the acclimatization phase, the health status of all pigs was evaluated by veterinary examination and confirmed to be free of ASFV by using a VIRTOTYPE^®^Real-time PCR kit (Indical Bioscience GMBH, Leipzig, Germany).

The vaccinated group was immunized first with 10^3^ TCID50/pig vaccine virus IM, and 14 dpv, they were again immunized with 10^3^ TCID50/pig. Whole blood and serum samples were collected on the days of vaccination. Four weeks after the boost immunization, pigs were challenged with 10^2^ HAD50 of virulent ASFV, Arm/07/CBM/c2 IM.

### 2.17. Samples Collection and Assessment of Clinical Signs

Evaluation of clinical signs was assessed on a daily basis beginning from 0 dpv to 56 dpv (28 dpc). Whole blood and serum samples were collected on the 0, 4, 7, 14, 21, and 28 dpv and then at 4, 7, 14, 21, and 28 dpc.

To assess virus load and shedding, rectal and oral swabs were collected at the same time points as blood and sera collection. Collected swabs were placed into tubes containing 1 mL of phosphate-buffered saline (PBS), vortexed, and incubated (10 min, RT). Blood was collected in plastic tubes containing K2-EDTA. Blood for serum was collected in serum separator tubes and centrifuged (1800 G, RT); sera were kept frozen at −20 °C until they were analyzed for anti-ASFV antibodies presence.

A complete necropsy was done on each animal after death or euthanasia. Tissue samples (i.e., the spleen, liver, kidneys, lungs, submandibular lymph nodes, tonsils, and bone marrow) were collected in 2 mL tubes. About 10% dilution in PBS (*w*/*v*) of each tissue was done by homogenization in TissueLyser^®^ (Qiagen, Germany).

### 2.18. ASFV Real-Time qPCR

A total of 200 µL of each sample (i.e., oral and rectal swabs, blood (1:10 *v*/*v* PBS), tissue homogenates) was used for DNA extraction. Manual column extraction was performed according to the Qiagen DNA Mini Kit protocol (Qiagen, Germany). Real-time PCR was conducted according to the VIROTYPE^®^ (Qiagen, Germany) manufacturer’s manual using the Rotor-Gene^®^ Q thermocycler (Qiagen, Germany).

A standard curve presenting dependence between Cq and virus titer was prepared (Microsoft Excel, Windows) as described previously [46]. Relevant Cq values obtained during the experiment were estimated and expressed as virus titers.

### 2.19. ELISA Assay

The serological status of the serum samples was determined by enzyme-linked immunosorbent assay (ELISA) with ID Screen^®^ African Swine Fever Indirect (IDVet Innovative diagnostic, Grabels, France) according to manufacturer instructions.

### 2.20. Statistical Analysis

The analyzed data were compared between groups using an unpaired *t*-test in GraphPad Prism (version 8.4.3, GraphPad Software). Statistically significant change was considered when *p* < 0.05.

## 3. Results

### 3.1. Ectopic Expression of the ASFV Protein pMGF505-2R Decreases IFN-β Production

The gene coding for the pMGF505-2R protein is present in the genome of the virulent strain Arm/07/CBM/c2 (LR812933.1) but absent in naturally attenuated strains, such as NH/P68 (KM262845.1). In addition, it displays a putative interaction domain with the STING protein [36]. Because IFN-β is the final product of the cGAS-STING signaling pathway, we first explored the effect of pMGF505-2R viral protein expression on IFN-β promoter activity. For this, we performed a luciferase assay that determines IFN-β promoter activity. To perform this assay, HEK/293T cells were transfected with (i) an empty vector, or TBK1 expression vector, to induce IFN-β production; (ii) either an empty vector or the MGF505-2R expression vector; (iii) the pIFNβ_Luc vector, which expresses luciferase under the IFN-β promoter; and (iv) the pRLTK vector, which expresses Renilla to normalize luciferase values. As shown in Figure 1A, TBK1 expression induces IFN-β promoter activation, whereas the expression of pMGF505-2R reduces such promoter activation in HEK/293T cells.

### 3.2. pMGF505-2R Modulates TBK1 Phosphorylation

To determine the point of the cGAS-STING pathway affected by the pMGF505-2R expression, we analyzed the effect of pMGF505-2R expression on TBK1, a key enzyme for cGAS-STING pathway activation. For this, BSRT7 cells were infected with MVA-T7 to induce activation of the cGAS-STING pathway and then transfected with the pIRES_MGF505-2R-myc expression vector or an empty vector. Figure 1B shows, on the one hand, that infection with MVA-T7 results in induction of TBK1 phosphorylation and that pMGF505-2R expression results in a decrease of the MVA-T7-induced TBK1 phosphorylation. These data demonstrate that pMGF505-2R expression is able to inactivate a central step of the cGAS-STING signaling pathway, in agreement with the above-observed inhibition of IFN-β promoter activation (Figure 1A).

### 3.3. pMGF505-2R Interacts with STING

As discussed in the Introduction, the HPV E7 and the Adenovirus E1A oncoproteins interact with STING through the “LxCxE” motif to inhibit cytoplasmic DNA-induced IFN-β signaling [36]. A sequence analysis of pMGF505-2R identified the presence of this motif at the C-terminal end of the ASFV protein (Figure 2A). Since we have shown that ectopic expression of pMGF505-2R is sufficient to affect the cGAS-STING signaling pathway and decrease IFN-β promoter expression, the next point was to study the possible interaction between the pMGF505-2R viral protein and STING, as a mechanism of this function.

To perform the co-immunoprecipitation assay, COS-1 cells transfected for 24 h with STING-Flag and pCAGGS_IRES_MGF505-2R-myc were used. Figure 2B shows how, after immunoprecipitating STING, a band corresponding to pMGF505-2R is observed after revealing with anti-myc, suggesting that pMGF505-2R interacts with STING under these conditions.

### 3.4. Generation of Recombinant Arm/07-ΔMGF505-2R-GFP Virus

The results analyzed so far suggest that pMGF505-2R may play a role in modulating the innate immune response, specifically through the cGAS-STING signaling pathway, possibly through interaction with STING. Bearing in mind that all these data are based on ectopic experiments, it is of great importance to further assess these results during ASFV infection to confirm them in a more complex scenario. For this reason, recombinant Arm/07-ΔMGF505-2R-GFP virus was generated from the parental Arm/07/CBM/c2 virus [37]. It should provide a concise and precise description of the experimental results, their interpretation, as well as the experimental conclusions that can be drawn.

To generate the recombinant Arm/07-ΔMGF505-2R-GFP virus, in-house-adapted CRISPR-Cas9 technology was used in COS-1 cells [34]. For this purpose, specific vectors, detailed in Materials and Methods, were used to transfect COS-1 cells, and after 24 hpt the transfected cells were selected with puromycin and infected with the parental Arm/07/CBM/c2 virus. After several rounds of purification, recombinant Arm/07-ΔMGF505-2R-GFP virus was obtained, in which the MGF505-2R gene was replaced by the GFP gene under the ASFV p72 promoter (see Figure 3A).

The complete genome sequence of Arm/07-ΔMGF505-2R-GFP was analyzed using Illumina (MicrobesNG) from purified DNA obtained from extracellular viral particles [37], with a high average coverage of 4460×. As seen in Figure 3B, a complete drop in reads in the region of the genome corresponding to the MGF505-2R gene (33872 bp-35452 bp) was found, confirming the deletion of the MGF505-2R gene, and the absence of other deletions in any other region of the genome was also assessed.

Genetic variability was analyzed using variant calling using GATK software, indicating both the number of single nucleotide polymorphisms (SNPs) and insertions/deletions (InDels). Analysis of the genetic variability of Arm/07-ΔMGF505-2R-GFP showed a very low number of mutations (Table 1): three InDels and three SNPs, which were confirmed by Sanger sequencing. A single cytosine insertion is present at genome position 14,005, in a cytosine-rich region (Poly-C), where a single cytosine insertion appears, affecting the MGF110-14L and MGF110-11L genes, causing a lengthening of 7 amino acids in the C-terminal region and a loss of 6 amino acids in the N-terminal region, respectively. The other two InDels are found in an intergenic region, while the SNPs affect three genes, causing the change of one amino acid for another in all three cases, affecting the MGF505-7R gene (Gly63Gln), the EP424R gene (Tyr307His), and the CP530R gene (Ser169Leu).

### 3.5. Growth Kinetic of Recombinant Arm/07-ΔMGF505-2R-GFP Virus

To test whether the absence of the MGF505-2R gene had an effect on the viral cycle, the in vitro growth kinetic was analyzed, compared to the parental virus growth kinetics. For this purpose, PAMs were infected with Arm/07/CBM/c2 or Arm/07-ΔMGF505-2R-GFP and at different times post-infection (0, 24, 48, and 72 hpi), total extracts were collected and titrated by TCID50. The results reflected in Figure 3C show that the absence of the MGF505-2R gene has no effect on the growth kinetics of the recombinant virus. In fact, from 24 hpi the Arm/07-ΔMGF505-2R-GFP mutant is produced at a higher ratio than the parental virus.

### 3.6. Arm/07-ΔMGF505-2R-GFP Induces IFN-β Production during Infection

Next, in order to study the effect of pMGF505-2R on IFN-β production during ASFV infection, we analyzed IFN-β production during infection with recombinant Arm/07-ΔMGF505-2R virus compared to infection with the virulent Arm/07/CBM/c2 strain and the attenuated NH/P68 strain. PAMs were infected for 16h with Arm/07/CBM/c2 or NH/P68, which respectively inhibits or induces IFN-β [11], or with the recombinant Arm/07-ΔMGF505-2R-GFP virus. In these conditions, IFN-β mRNA levels were analyzed with RT-qPCR. As shown in Figure 4A, and interestingly, both during infection with Arm/07-ΔMGF505-2R-GFP and NH/P68, higher IFN-β mRNA levels were produced, whereas during infection with the virulent Arm/07/CBM/c2 there was almost no IFN-β mRNA production. As a control, the levels of infections of CP204L mRNA were assessed (Figure 4B), and MGF505-2R mRNA levels were also analyzed to confirm the lack of MGF505-2R expression in both NH/P68 and Arm/07-ΔMGF505-2R-GFP (Figure 4C). These results confirm the data obtained in ectopic MGF505-2R expression experiments and show a clear role of pMGF505-2R in controlling IFN-β production during ASFV infection.

### 3.7. Infection with Arm/07-ΔMGF505-2R-GFP Does Not Inhibit Phosphorylation of cGAS-STING Pathway Components

Once it was determined that Arm/07-ΔMGF505-2R-GFP induces IFN-β production in PAMs and that, therefore, MGF505-2R gene is partly responsible for the ability to inhibit IFN-β production, we wanted to determine whether this inhibition was mediated through the cGAS-STING pathway. For this purpose, we analyzed the phosphorylation status of several of the main proteins of this pathway (STING, IRF3, and TBK1) at different post-infection times (4, 8, and 16 hpi) during infection with the recombinant virus compared to the parental Arm/07/CBM/c2 virus. The results shown in Figure 4D indicate that infection with the parental Arm/07/CBM/c2 virus barely induces phosphorylation of TBK1, IRF3, and STING at 4 hpi, and this phosphorylation decreases as the infection progresses, as we previously described [11]. The total levels of STING and TBK1 remain constant, while those of IRF3 could not be tested. In contrast, phosphorylation of these proteins during Arm/07-ΔMGF505-2R-GFP infection was higher at each of the times analyzed, indicating activation of these elements during Arm/07-ΔMGF505-2R-GFP infection. These results again point to the MGF505-2R gene as a modulator of the cGAS-STING pathway and, importantly, demonstrate that, in the absence of this gene, the parental Arm/07/CBM/c2 partly loses the ability to counteract the activation of the cGAS-STING pathway.

Altogether, our results indicate that the viral pMGF505-2R protein is involved in the modulation of the cGAS-STING signaling pathway and in the inhibition of IFN-β production, and consequently, the Arm/07-ΔMGF505-2R-GFP activates the innate immune response during infection.

Effect of infection with Arm/07-ΔMGFR505-2R-GFP in vivo: Experiment I.

A first animal experiment was conducted in collaboration with Careside Co., Ltd., under BSL3 biosafety conditions at the Korea Zoonosis Research Institute, Jeonbuk National University. A total of eight animals were divided into two experimental groups: a first unvaccinated control group (n = 4) and a second group immunized with Arm/07-ΔMGFR505-2R-GFP (n = 4).

### 3.8. Arm/07-ΔMGF505-2R-GFP Is Attenuated In Vivo to a Dose of 10^2^ TCID50

Vaccination with Arm/07-ΔMGF505-2R-GFP was performed intramuscularly at 10^2^ TCID50/animal dose. During 28 days post-vaccination, the evaluation factors for safety, such as body temperature, body weight, and clinical signs of African swine fever (ASF) disease, lethargy, and digestive problems were observed, and blood and swab samples were collected and analyzed. No vaccinated pigs showed obvious signs of disease, and as shown in Figure 5A, the body temperature of the animals remained stable and below what was considered a fever (41 °C). When the appearance of clinical signs of ASF was analyzed according to the table of values established in [44], where the maximum value is 40, the mean of the four animals in the clinical signs score did not exceed 3 (Appendix A). These data reveal that the recombinant Arm/07-ΔMGF505-2R-GFP virus is attenuated in vivo.

### 3.9. Arm/07-ΔMGF505-2R-GFP Partially Protects against the Circulating Virulent Strain ASFV/Korea/pig/PaJu1/2019

Once the in vivo attenuation of Arm/07-ΔMGF505-2R-GFP was determined, we proceeded to analyze its efficacy in protecting against a circulating virulent strain, the genotype II, ASFV/Korea/pig/PaJu1/2019. For this purpose, after 28 days post-vaccination, animals from the control and vaccinated groups were challenged intramuscularly at a dose of 10^2^ TCID50/animal.

As seen in Figure 5C, all unvaccinated control animals died before 14 dpc, whereas only one of the vaccinated animals succumbed at 10 dpc. These results show that 75% of the animals vaccinated with Arm/07-ΔMGF505-2R-GFP survived the challenge. Consistent with the protection data, the body temperature of the immunized animals remained below 41 °C except for two animals between 6 and 8 dpc, of which one reverted (animal 63), while animal 64 finally died at 10 dpc (Figure 5B).

On the other hand, viremia was analyzed in both immunized and non-immunized animals before and after the challenge. As shown in Table 2, in vaccinated animals, before the challenge, viremia was detected in only one animal (#61) at 5 dpv, and remained undetectable in the rest of the animals. After the challenge, high levels of viremia were detected in the unvaccinated animals (control) and in animal #64 of the vaccinated group from day 5 dpc. The viremia detected in these animals is consistent with survival since all these animals, which had high levels of viremia in blood after challenge, died in the following days (Figure 5B,C). The rest of the vaccinated animals had only mild levels of viremia after the challenge.

### 3.10. Activation of Humoral Response after Vaccination with Arm/07-ΔMGF505-2R-GFP

To determine the activation of the adaptive immune response in animals vaccinated with Arm/07-ΔMGF505-2R-GFP, the production of specific antibodies against ASFV was analyzed. For this purpose, these antibodies were detected in the serum of immunized animals every 7 days from day 0 to 28 dpv and from day 0 to 28 dpc (49 dpv). Animal sera were assayed in a specific ELISA for antibodies against the ASFV viral protein p72 (encoded by B646L) (INgezim PPA COMPAC, Ingenase). As shown in Figure 5D, a high level, around 70%, of virus-specific antibodies is detected from day 14 dpv and maintained at 21 dpc in immunized animals. As expected, in non-immunized animals, no antibodies were detected after the challenge since they died within 10 days.

Effect of infection with Arm/07-ΔMGFR505-2R-GFP in vivo: Experiment II.

In the previous animal experiment, we deduced that the Arm/07-ΔMGF505-2R vaccine candidate was fully attenuated and protected up to 75% against a virulent challenge at a dose of 10^2^ TCID50. In order to increase the efficacy of protection, while maintaining safety, a second animal study was performed in which the experimental conditions were altered, immunizing with a first-boost protocol at a dose of 10^3^ TCID/50. This second study was conducted at the National Veterinary Research Institute (NVRI) in Pulawi, Poland.

### 3.11. Arm/07-ΔMGF505-2R-GFP Is Attenuated In Vivo Using a Prime-Boost Protocol at a Dose of 10^3^ TCID50

After immunization, no clinical abnormalities were observed between 0 and 28 dpv. Animals have presented normal behavior and rectal temperature, except one pig (pig D, Δ2R group) with 2 days of transient fever at 14 and 15 dpv (Figure 6A). The genetic material of ASFV was not detected in any samples (i.e., swabs, blood) before the challenge.

### 3.12. Arm/07-ΔMGF505-2R-GFP Partially Protects against the Virulent Strain Arm/07/CBM/c2

In the control non-vaccinated group, mortality reached 100% until 6 dpc (Figure 6B,C). The mean incubation period was estimated as 3 days. At 5 dpc, two animals were found dead, and one animal was euthanized due to reaching the humane endpoint. In contrast, in Arm/07-ΔMGF505-2R-GFP-vaccinated animals, although fever, apathy, and mild ataxia were noticeable, the mean incubation period was longer than in group WT and estimated as 5 (±2) days. Rectal temperature showed a tendency to decrease with a single case of returning fever caused by bacterial joint co-infection (Figure 6B). One animal (pig C) was found dead in 9 dpc, and one pig (pig A) required euthanasia due to dyspnea (15 dpc) (Figure 6C). In this group, three our of five animals survived till the end of the experiment, with two pigs showing a complete lack of negative outcomes (one pig had a joint infection).

Regarding detection of ASFV genetic material, at 4 dpc, virus DNA was detectable in all matrices (rectal and oral swabs, blood) among animals in the non-vaccinated group. In the same sampling timepoint, significantly higher Ct values in blood were observed in the group of vaccinated animals (*p* < 0.0001) (Figure 7A). Between 4 and 7 dpc, the evolution of the viremia among vaccinated animals showed evidence of virus replication. Subsequently, between 7 and 28 dpc, the virus clearance tendency in blood was noticeable, leading to a decrease (pig D) or even removal of ASFV genetic material (28 dpc, pig E). Until the end of the experiment, blood samples of two pigs were found to be PCR negative (pig B—all blood samples negative; pig E—negative at 28 dpc).

Reduced shedding in oral and rectal swabs was recorded in the vaccinated group. The genetic material of ASFV was found in only one pig’s (pig A) oral swabs (at 35 and 42 dpc—Figure 7B), while other oral swabs and all rectal swabs from this group were found to be PCR negative (Figure 7C).

Typically for ASF, macroscopic lesions (i.e., splenomegaly, hemorrhagic lymphadenitis, and hyperemic tonsils accompanied by a high amount of exudative fluid found in the abdomen or pleural cavity) were observed during necropsy in all animals from the non-vaccinated group. Only milder lesions were found in animals that survived the challenge from the vaccinated group; however, in this group, splenomegaly and enlargement of submandibular lymph nodes were noticeable. One pig (pig C, found dead at 9 dpc) from this group presented lesions typical for acute ASF, while one pig presented no lesions at all (pig B). The frequency of lesions is presented in Table 3.

A lower virus load was noticeable in samples collected from the vaccinated group during necropsy, out of which some internal organs (i.e., spleen, kidney, liver, lymph nodes, and bone marrow) presented statistically significant lower virus load (spleen, *p* = 0.0005, kidney *p* = 0.0006, liver *p* = 0.010, lymph nodes *p* = 0.014, bone marrow *p* = 0.035), as shown in Figure 8A.

### 3.13. Humoral Response Is Activated after Vaccination with Arm/07-ΔMGF505-2R-GFP

Again, to determine the activation of the adaptive immune response in animals vaccinated with Arm/07-ΔMGF505-2R-GFP in this new animal experiment, the production of specific antibodies against ASFV was analyzed. A visible tendency of raised OD values was found in the vaccinated group, beginning at 7 dpv. At 28 dpv (day of the challenge), all vaccinated pigs presented elevated OD values—mean OD values in vaccinated animals were estimated as 0.41 (±0.2) vs. 0.07 (±0.0) in the control group (Figure 8B). However, according to ELISA protocol, three out of five of the vaccinated animals were found to be seropositive (pig B,E)/doubtful seropositive (pig D) on the challenge day. At 35 dpv, all animals (five out of five) from the vaccinated group were found to be seropositive and presented raised antibody titers (Figure 8B). In comparison, all challenged animals from the WT group were found to be seronegative, with no elevated OD values.

## 4. Discussion

ASFV is currently the cause of the largest animal pandemic, affecting domestic pigs and wild boar. The disease is widespread in more than 40 countries across four continents [8], enormously affecting the meat industry and the food balance in affected countries. In the absence of a commercial vaccine, both in affected areas and preventively in non-endemic areas, the only measures used so far against the ASFV spread are early diagnosis, regionalization, and culling of infected animals, measures that are insufficient in view of the continued spread of the virus.

A crucial topic, both fundamental for understanding the biology of the virus and applied to the development of new vaccines and tools against ASF, is the study of the molecular basis of ASFV virulence. In this regard, our group described a few years ago the importance of the control of type I IFN expression, and in particular of the control of the cGAS/STING pathway in ASFV virulence [11]. Briefly, the cGAS/STING pathway is activated upon detection of DNA in the cell cytoplasm by the cGAS sensor, which triggers the generation of cGAMP that binds and activates STING. STING traffics to the Golgi, where it recruits TBK-1, which autophosphorylates and phosphorylates STING, resulting in IRF3 phosphorylation and translocation to the nucleus, acting as a transcription factor for IFN-β expression [47]. Virulent but not attenuated ASFV strains are able to counteract the cGAS/STING pathway and type I IFN production [11], and, in fact, multiple ASFV genes have been subsequently identified in counteracting multiple the elements of this pathway, highlighting the importance of the cGAS/STING pathway in ASFV infection, though many of them are only based on in vitro results by ectopic expression of the viral protein.

Some of these viral proteins seem to affect the first steps of the pathway, such as the QP383R that interacts with cGAS, preventing DNA binding [13] or C129R and EP364R that degrade cGAMP [12]. Other ASFV proteins counteract TBK1 action by inducing its degradation, as A137R [18], MGF-110-9L [17], MGF360-11L or MGF505-7R do, the two latter further interacting with IRF7 [48,49]; while others degrade IRF3, such as M1249L or MGF360-14L [50,51] or inhibit its activation, such as E120R, E301R or S137R [52,53,54]. Multiple ASFV proteins seem to interact with STING, leading to counteract its activation, such as B175L, D117L (P17), MGF-505-7R, MGF505-11R, EP402R (CD2v) or L83L [14,15,16,19,55,56]. One of those, as we described here and further discussed later, is MGF505-2R. This gene had previously been identified as an inhibitor of the innate immune response by generating a CN/GS/2018-ΔMGF505-2R mutant, described to be able to induce a number of IFN-related, inflammatory, and immune response genes, although the mechanisms of this function remained elusive [57].

In this work, we have demonstrated that MGF505-2R, which is a late-expressed gene during the viral cycle, is able to counteract the activation of the cGAS/STING pathway and IFN-β production during infection. A recombinant Arm-ΔMGF505-2R virus was generated to precisely analyze the effect of deletion of this gene in the context of ASFV infection, and it was genetically characterized by NGS, confirming the absence of this gene alone in the virus genome, as well as the absence of off-target mutations, except for two point mutations in intergenic regions, and three SNPs in the MGF505-7R, EP424R and CP530R genes (Table 1). MGF505-7R has also been implicated in the control of IFN-β [58], so this SNP could also have contributed to the lack of control over IFN-β. However, further studies are needed to corroborate this hypothesis, such as determining the specific sites of MGF505-7R involved in this function.

The mechanism by which pMGF505-2R could exert control over the cGAS-STING pathway could involve STING interaction. In addition, we have identified a LDCLE motif in MGF505-2R, analogous to that found in E7 from human papillomavirus (HPV) and E1A from adenovirus, which has also been shown to bind to STING and counteract the cGAS/STING pathway [36].

We first hypothesized that the absence of this protein was linked to virulence in domestic pigs, similar to other ASFV proteins involved in the control of the cGAS/STING pathway, which have been found to be directly related to virulence. Examples of that would be ASFV MGF-505-7R [15], A137R [18,33], or MGF-110-9L [17,59].

To prove this hypothesis, and once we had identified pMGF505-2R as a cGAS/STING counteractor preventing IFN-β production, we performed a first animal trial where we immunized with a single dose of 10^2^ TCID50/animal with the recombinant Arm-ΔMGF505-2R, and after 14 days post-immunization we proceeded to a challenge with the virulent Korean genotype II circulating strain ASF/Korea/Pig/Paju/2019 at a dose of 10^2^ HAD/animal, which results in a 75% survival. It would be interesting to analyze in future animal experiments whether the attenuation observed during immunization with this recombinant virus is related to an increase in IFN-β production, correlating with the data observed in vitro. In addition, other factors might play a role in this attenuation since other genes related to inflammation and innate immune response have been found to be upregulated in a ΔMGF505-2R mutant [57]. In order to try to increase protection while preserving safety, we considered modifying the immunization protocol and increasing the dose. Therefore, we performed a new in vivo experiment in which animals were immunized following a prime-boost protocol, with two doses of 10^3^ TCID50/animal separated by 14 days, revealing once again the complete safety of the LAV prototype. However, no increase in survival was found when vaccinated animals were challenged with the virulent genotype II Arm/07/CBM/c2 strain, as the prototype surprisingly showed only a rate of around 60% in this last trial. Although we observed lower survival in Experiment II, the efficacy results should be considered similar. Firstly, because the vaccinated groups differed in the number of pigs (4 vs. 5), and secondly because in Experiment II, in one of the two dead pigs, which was euthanized, co-infection with *Streptococcus suis* was diagnosed, which probably had a synergistic impact on the course of the disease.

## 5. Conclusions

In conclusion, our work identified the MGF505-2R gene as an ASFV immunomodulator gene involved in the control of the innate immune response through the control of the cGAS/STING pathway, whose deletion causes full attenuation of the virulent Arm/07/CBM/c2 strain, establishing a direct link between the control of the cGAS/STING pathway and ASFV virulence. Moreover, the recombinant Arm/07-ΔMGF505-2R-GFP virus behaved as a putative LAV prototype conferring around 60–70% protection and may be a good candidate for generating improved LAVs combining the deletion of MGF505-2R together with other gene candidates for protection, or, alternatively, by generating other vaccination tools from the backbone of this Arm/07-ΔMGF505-GFP prototype. It is becoming increasingly clear that safety is probably the main requirement of LAV vaccines against ASF. While protection is almost guaranteed by some of the commercial tools that appeared in the market during the pandemic, important safety concerns have arisen around them that should be solved before using these kinds of prototypes widely in the world.

## Figures and Tables

**Figure 1 vaccines-12-00407-f001:**
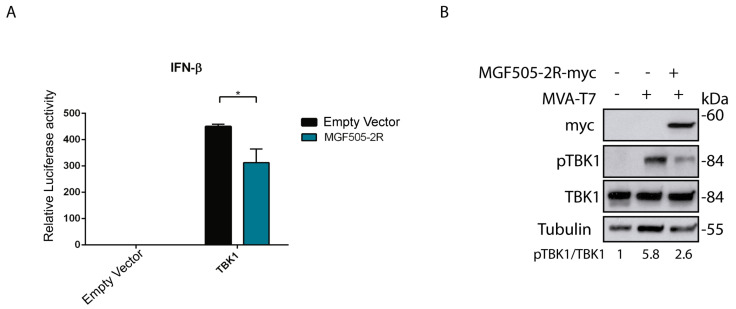
Ectopic expression of pMGF505-2R reduces IFN-β promoter activity and reduces TBK1 phosphorylation. (**A**) HEK/293T cells (120,000 cells) were transfected with pTBK1_Flag (50 ng), pCAGGS_IRES_MGF505-2Rmyc (1 µg), pIFNβ_Luc (50 ng), pRLTK (25 ng) and empty vector (to equal amount in ng) as indicated. At 24 h post-transfection (hpt), cells were lysed, and luciferase activity was analyzed using luciferase reporter assay after IFN-β promoter induction with pTBK1_Flag. The graph is representative of three different experiments. (**B**) BSRT7 cells were infected with MVA-T7, and 1 hpi cells were transfected with an empty vector or pIRES_MGF505-2R-myc vector (1000 ng). At 16 hpt, cells were lysed with RIPA. Phosphorylation status and total expression of proteins were analyzed with Western blot with anti-pTBK1, anti-TBK1-E8I3G, anti-tubulin, and anti-myc (MGF505-2R) antibodies. Data were statistically analyzed by using a Student’s *t*-test (* *p* < 0.05).

**Figure 2 vaccines-12-00407-f002:**
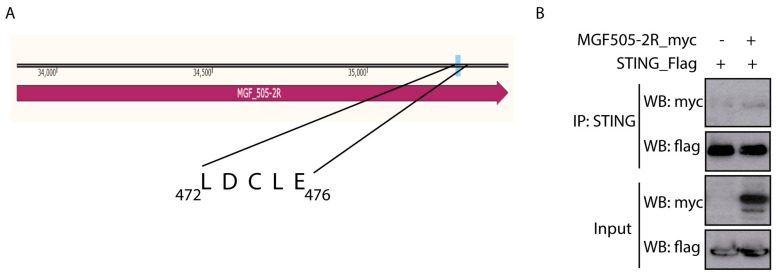
ASFV pMGF505-2R viral protein interacts with STING. (**A**) Diagram of the pMGF505-2R protein showing the location of the identified LxCxE motif (in particular, the LDCLE sequence). (**B**) COS cells were transfected with pSTING-Flag (0.5 µg/1-10^6^ cells) and pCAGGS_IRES_MGF505-2R-myc or empty vector (3 µg/1–10^6^ cells). Cells were lysed with lysis buffer from the IP/CO-IP kit (Pierce TM). A fraction of the total extract (input) was separated, and then STING was immunoprecipitated using an anti-STING antibody bound to magnetic beads. Inputs and immunoprecipitation (IP) samples were analyzed with WB using anti-myc (MGF505-2R-myc) and anti-Flag (STING-Flag) antibodies.

**Figure 3 vaccines-12-00407-f003:**
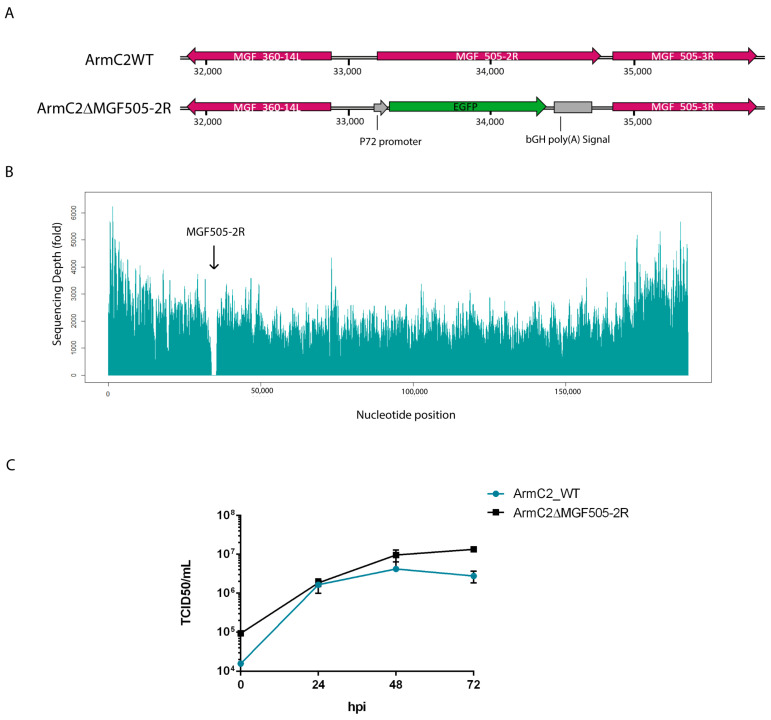
Generation and behavior assessment of Arm/07-ΔMGF505-2R-GFP recombinant virus. (**A**) Representative schematic of the genetic modification produced in the Arm/07/CBM/c2 genome. Replacement of the complete MGF505-2R ORF by the GFP gene under the viral p72 promoter. (**B**) Coverage plot of Illumina sequencing of Arm/07-ΔMGF505-2R-GFP genome with respect to the Arm/07/CBM/c2 sequence. (**C**) Comparison of growth kinetics of Arm/07-ΔMGF505-2R-GFP virus with Arm/07/CBM/c2 virus. PAMs were infected with a MOI = 0.5 of each virus. Infected cells were collected at 0, 24, 48, and 72 hpi, and viruses were titrated by TCID/50. Arm/07/CBM/c2 is represented in blue, and Arm/07-ΔMGF505-2R-GFP is represented in black.

**Figure 4 vaccines-12-00407-f004:**
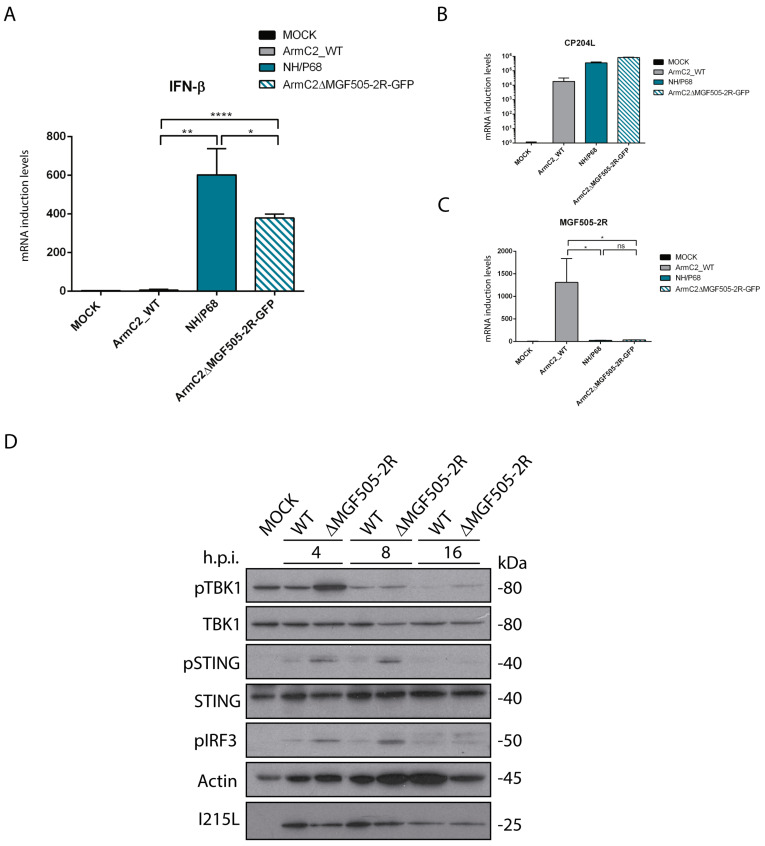
Arm/07-ΔMGF505-2R-GFP infection induces IFN-β expression and did not prevent cGAS-STING pathway activation in infected PAMs. PAMs were infected with NH/P68, Arm/07/CBM/c2, Arm/07-ΔMGF505-2R-GFP with a MOI = 4 or non-infected (Mock). Cells were harvested at 16 hpi and analyzed using RT-qPCR for IFN-β (**A**), viral CP204L (**B**), and viral MGF505-2R (**C**) mRNA levels (Mean ± S.E.M) (n = 3). Data were statistically analyzed using a Student’s *t*-test (* *p* < 0.05; ** *p* < 0.01; **** *p* < 0.0001). (**D**) PAMs were infected with Arm/07/CBM/c2 or Arm/07-ΔMGF505-2R-GFP with a MOI = 2 or not infected (Mock). Cells were lysed in RIPA buffer supplemented with phosphatase and protease inhibitors at 4, 8, and 16 hpi. Phosphorylation status and total expression of proteins of interest were analyzed using Western blot with anti-p-STING, anti-STING, anti-p-IRF3, anti-p-TBK1, anti-TBK1, anti-I215L, and anti-actin antibodies.

**Figure 5 vaccines-12-00407-f005:**
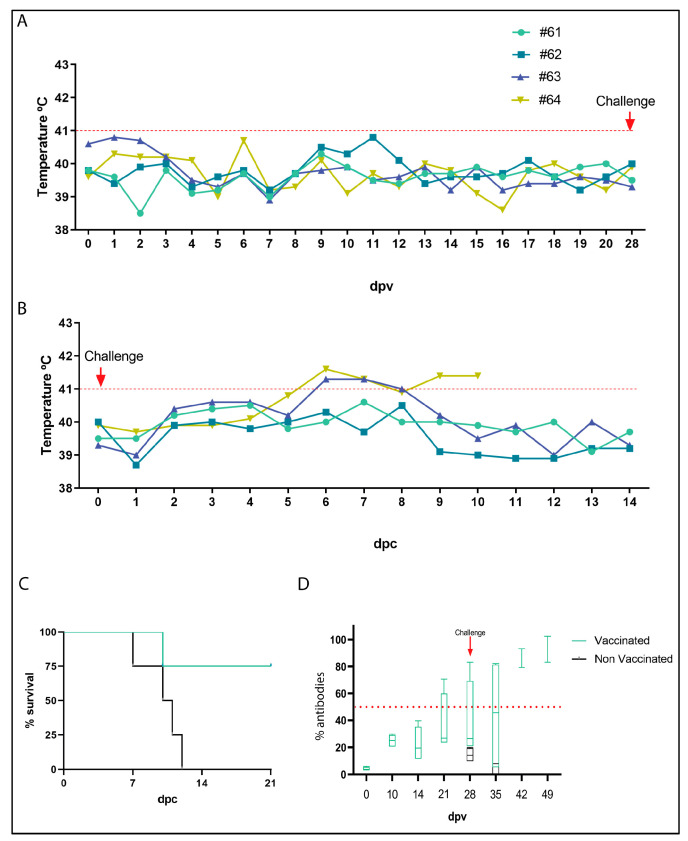
Arm/07-ΔMGF505-2R-GFP is attenuated in vivo and confers protection against the virulent Korean Paju 2019 strain challenge. (**A**) Body temperature of vaccinated animals 61, 62, 63, and 64 was measured from 0 to 28 days post-vaccination (dpv). The red dotted line marks the body temperature above which it was considered a fever in pigs (41 °C). (**B**) Body temperature of vaccinated animals with Arm/07-ΔMGF505-2R-GFP from 0 to 14 days post-challenge (dpc). The red arrow indicates the challenge day (28 dpv or 0 dpc). The red dashed line marks the body temperature above which fever is considered (41 °C). (**C**) Percent of survival of vaccinated animals immunized with Arm/07-ΔMGF505-2R-GFP (green) or not vaccinated (black) at 21 dpc. (**D**) Detection of antibodies in sera from animals vaccinated with Arm/07-ΔMGF505-2R-GFP. Mean percentage of ASFV-specific antibodies in serum samples from vaccinated animals (green) and unvaccinated animals (black) at 0, 7, 14, 21, 28, 28, 35, 42, and 49 dpv and 0, 7, 14, and 21 dpd (n = 4). The red arrow indicates the challenge day with the Korea/2019 strain (28 dpv or 0 dpc). The red dashed dotted line marks the minimum percentage of antibodies above which a positive value is considered (>50%).

**Figure 6 vaccines-12-00407-f006:**
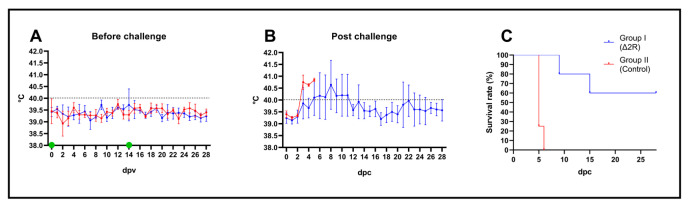
Arm/07-ΔMGF505-2R-GFP is attenuated in vivo with a 10^3^ dose in a prime-boost experiment and confers partial protection against virulent Arm/07/CBM/c2 strain challenge. (**A**) Rectal temperature of animals before challenge. Green dots indicate first and boost inoculation of Arm/07-ΔMGF505-2R group with 10^3^ TCID50 dose per animal. (**B**) rectal temperature of animal post-challenge. Black dashed lines indicate the fever threshold. (**C**) survival rate. Error bars indicate standard deviation.

**Figure 7 vaccines-12-00407-f007:**
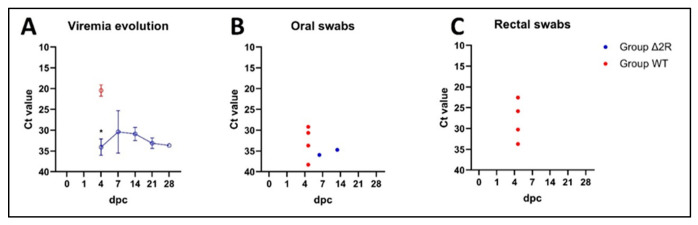
Viremia evolution and shedding pattern in swabs of control and vaccinated animals with Arm/07-ΔMGF505-2R-GFP. (**A**) Viremia evolution during the experiment (mean Ct values): initial virus load in the blood of vaccinated animals is significantly lower (*p* < 0.0001) than in the blood of WT group animals. Virus clearance tendency is noticeable between 7 and 28 dpc. (**B**) shedding pattern (individual values) in oral swabs. (**C**) shedding pattern (individual values) in rectal swabs. *—statistically significant, dpc—day post-challenge. Error bars indicate standard deviation.

**Figure 8 vaccines-12-00407-f008:**
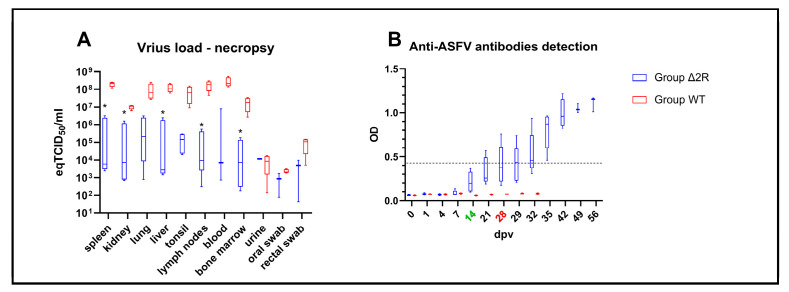
Virus load in tissues and antibodies detection from control or vaccinated animals with Arm/07-ΔMGF505-2R-GFP. (**A**) Virus load in internal organs and chosen samples collected during necropsy. (**B**) optical density values (OD) of anti-ASFV antibodies detection. The boxes represent the 50% between the 25 and 75% quartiles. The line inside the box indicates the median. The top and bottom lines denote maximum and minimum values. Dpv—days post-vaccination, green color—boost timepoint, red color—challenge timepoint, *—statistically significant, eq—equivalent, Black dashed line—mean negative cutoff threshold.

**Table 1 vaccines-12-00407-t001:** List of variants identified by variant calling analysis of Arm/07-ΔMGF505-2R-GFP compared to Arm/07/CBM/c2. The position corresponds to the position in the whole genome of parental Arm/07/CBM/c2.

Position	Type	Location	Mutation	Description
14,005	InDel	MGF110-11L MGF110-14L	A/AC	Loss of 6 aa N’Term Acquisition of 7 aa C’Term
15,444	InDel	Intergenic region	A/ACC	
17,403	InDel	Intergenic region	CG/C	
41,689	SNP	MGF505-7R	G/A	Gly63/Gln63
72,293	SNP	EP424R	T/C	Tyr307/His307
126,682	SNP	CP530R	C/T	Ser169/Leu169

**Table 2 vaccines-12-00407-t002:** Viremia of control and vaccinated animals with Arm/07-ΔMGF505-2R-GFP. The analysis of viremia in blood was performed by amplifying the viral gene B646L (p72) using real-time PCR. The unvaccinated animals (control) correspond to animals no. #65, #66, #67 and #68. The vaccinated animals correspond to animals no. #61, #62, #63 and #64. Representation of the viremia degree: no viremia (>37) in black; mild viremia (37–35) in pastel orange; moderate viremia (35–30) in orange; and high viremia (<30) in red.

Animal	0 dpv	3 dpv	5 dpv	7 dpv	10 dpv	14 dpv	21 dpv	0 dpc	3 dpc	5 dpc	7 dpc	10 dpc	14 dpc	21 dpc
#61	38.2	ND	38.4	37.9	35.5	35.1	37.2	ND	30.7	36.9	37.1	37.9	30.9	34.1
#62	ND	ND	ND	ND	ND	ND	ND	ND	ND	37.1	38.1	38.4	37.2	39.3
#63	ND	ND	ND	ND	39.0	ND	ND	35.0	38.3	30.4	31.3	33.6	33.7	37.3
#64	ND	ND	ND	ND	ND	ND	ND	ND	36.9	23.6	20.6	19.4		
#65								ND	ND	ND	19.8	18.2		
#66								ND	ND	20.8	17.4	17.7		
#67								ND	ND	19.3	17.1			
#68									39.3	36.5	22.9			

**Table 3 vaccines-12-00407-t003:** Frequency of lesions observed in necropsy. The numbers represent the number of animals with detected lesions.

*Lesion*	Arm/07-ΔMGF505-2R: n = 5	Frequency	Arm/07/CBM/c2: n = 4	Frequency
Splenomegaly	3	60%	4	100%
Hyperaemia and/or enlargement of lymph nodes *	4	80%	4	100%
Abdomen—exudative fluid	1	20%	4	100%
Hyperaemia of tonsil	2	40%	4	100%
Petechiae in kidneys	1	20%	1	25%
Pleural exudative fluid	1	20%	2	50%
Hyperaemia of lungs	1	20%	2	50%
Nasal discharge	1	20%	0	0%
Pericardial exudative fluid	2	40%	3	75%

*—submandibular, gastrohepatic or mesenteric.

## Data Availability

The original contributions presented in the study are included in the article/Appendix A, further inquiries can be directed to the corresponding author/s.

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
