# Peer review of "Deletion of MGF505-2R Gene Activates the cGAS-STING Pathway Leading to Attenuation and Protection against Virulent African Swine Fever Virus"

_vaccines, 2024, doi:10.3390/vaccines12040407_

Round 1

Reviewer 1 Report

Comments and Suggestions for Authors

Reviewer’s suggestion

African swine fever virus (ASFV) causes a highly lethal swine disease worldwide, severely affecting the pig industry. At present, there is no fully safe commercial vaccine. Therefore, the identification of the virulence-related genes of ASFV and further development of live attenuated vaccines (LAVs) are particularly important. In this study, the authors identified MGF505-2R as an ASFV immunomodulator gene involved in the inhibition of IFN-β production through cGAS/STING pathway, and the deletion of MGF505-2R caused attenuation of the virulent Arm/07/CBM/c2 strain. In vitro, pMGF505-2R interacted with STING to control IFN-β production, and the mRNA level of IFN-β in PAM infected with Arm/07-ΔMGF505-2R-GFP was higher than that in PAM infected with WT. Arm/07-ΔMGF505-2R-GFP is attenuated in vivo, and immunization with Arm/07-ΔMGF505-2R induced the generation of antibodies and confers partially protection against virulent ASFV strains challenge. This is a meaningful study and provides a gene candidate for generating improved LAVs.

Major questions

1.         ASFV pMGF505-7R plays a crucial role in inhibition of IFN-β production (Li et al., 2021). The analyzed results of the complete genome sequence of Arm/07-ΔMGF505-2R-GFP showed the mutation at the MGF505-7R gene (Gly63Gln). Whether the mutant has affected the function related with IFN-I signaling of MGF505-7R?

2.         Why lower survival when vaccinated animals were challenged with the parental stain compared to the circulating stain ASF/Korea/Pig/Paju/2019? Please address the question in the Discussion section.

3.         In vivo, whether Arm/07-ΔMGF505-2R-GFP infection results higher level of IFN-β compared to ASFV-WT infection? Huang et al. reported that ASFV-Δ2R induced the most dramatic expression of interferon-related genes and inflammatory and innate immune genes. Is the reduced virulence of Arm/07-ΔMGF505-2R-GFP related to other factors? Please discussed Need to discuss.

4.         In the Figure 1 and 4, protein expression of IRF3 was not showed. Line 583-584, the inaccurate expression “Infection with Arm/07-ΔMGF505-2R-GFP induces IFN-β expression and did not prevent cGAS-STING pathway activation in infected PAM” should be corrected for “Arm/07-ΔMGF505-2R-GFP infection induces higher expression of IFN-β and activates cGAS-STING pathway more strongly” or similar meaning expressions.

Minor questions

1.       Line 400, Arm/07/CBM/2” corrects to “Arm/07/CBM/c2”.

2.       Line 466, IRF3 phosphorylation was not detected in the Figure 1. “IRF3 and” should be deleted.

3.       In the manuscript, the MGF505-2R protein should be represented by “pMGF505-2R”.

4.       Line 669, there is a mistake in “the viral gene B602L (p72)”.

5.       Line 810, there is an extra space between ”generated to”.

6.       In the Table S1, note the cells merge in “Skin Haemorrhage”.

Reviewer 2 Report

Comments and Suggestions for Authors

In this study,the authors identified that the MGF505-2R gene acted as an inhibitor of the cGAS-STING pathway and decreased IFN-βproduction by interacting with the STING protein.In addition,immunization of a recombinant virus lacking Arm/07-ΔMGF505-2R, which resulted in a completely attenuation.Finally, immunization with Arm/07-ΔMGF505-2R induced the generation of antibodies and proved to be partially protective against a challenge against virulent ASFV strains.Overall,the study experiments are well designed,the results are strong,the conclusions are reliable,the article has certain significance.However,there are still the following points need to further improvement.

1.In Figure 1,the authors mentioned in the legend of Figure 1 that ectopic expression of MGF505-2R can reduce IRF3 phosphorylation, but it is not shown in the figure.Please ask the author to correct it.

2.In Figure 2,Please check the flag image in the Input diagram.

3.Line 453,The sentence is grammatically incorrect and lacks a subject.

4.Please verify the format of the eighth item in the reference.

5.Please verify the number callout of 14dpv in #62 in Table 2

Comments on the Quality of English Language

There are minor errors in English grammar that need to be corrected.

Reviewer 3 Report

Comments and Suggestions for Authors

In the manuscript of Sunwoo et al., the authors demonstrated the role of the MGF505-2R gene of ASFV in the inhibition of the cGAS/STING pathway through interaction with the STING protein. In addition, immunization of pigs with a recombinant virus lacking this gene (Arm/07-ΔMGF505-2R), showed its complete attenuation and partial protection against the virulent ASFV strain.

1. Objectives: the objectives and the rationale of the study clearly stated.

2.  Methods: The methods are described in sufficient detail to understand the approach used.

3.  Results: the results are clearly presented.

4.  Interpretation: Conclusions are supported by the obtained data.

5.  Other comments:

Lines 1-4 – The title of the manuscript has to be changed. My suggestion is: The MGF505-2R gene-deleted recombinant ASF virus is attenuated, does not inhibit the cGAS-STING pathway,  and partially protects against the virulent African swine fever virus challenge.

Line 25 – Vietnam became the 1st country to authorise commercial use of the ASF MLV vaccine. Partners from 5 Asian countries have signed contracts to purchase this vaccine. So, regulatory authorities of these countries consider this vaccine as a safe enough.

Line 388 – Meaning of “The positive blocking % was >50%” is not clear. According to the INGEZIM PPA Compac booklet, “Two cut offs are used for the results interpretation. Samples will be considered positive if their OD value is equal to or lower than the positive cut off. Samples will be considered negative if their OD value is equal to or higher than the negative cut off. Samples with OD value between both cut offs will be considered doubtful.” Please interpret ELISA results accordingly to the manufacturer’s instructions.

Line 466 – IRF-3 phosphorylation is not shown in this figure.

Line 467 – Meaning of “cell/condition” is not clear.

Line 539 – Refer to the Figure 3B, not 4B.

Figure 4D – Total IRF3 is missing in this figure, only pIRF3 is shown.

Line 653 – Refer only to the Figure 5C, not 5B.

Figure 5D – Present ELISA data in a similar way as they are presented on Figure 8.

Line 678 – Change “dpd” to “dpc”.

Line 683 – Clarify that non-vaccinated and challenged animal did not developed antibodies because they died within 10 days.

Comments on the Quality of English Language

Minor editing of English is required. I found few sentences with grammar errors.    
